# Non-Invasive Brain Stimulation and Sex/Polypeptide Hormones in Reciprocal Interactions: A Systematic Review

**DOI:** 10.3390/biomedicines11071981

**Published:** 2023-07-13

**Authors:** Jitka Veldema

**Affiliations:** Department of Sport Science, Bielefeld University, 33501 Bielefeld, Germany; jitka.veldema@uni-bielefeld.de

**Keywords:** non-invasive brain stimulation, (repetitive) transcranial magnetic stimulation, direct current stimulation, individualised medicine, sex hormones, polypeptide hormones, reciprocal interactions, oestradiol, progesterone, androgen, testosterone, luteinizing hormone, follicle-stimulating hormone, gonadotropin-releasing hormone, dehydroepiandrosterone

## Abstract

A better understanding of interindividual differences and the development of targeted therapies is one of the major challenges of modern medicine. The sex of a person plays a crucial role in this regard. This systematic review aimed to summarise and analyse available evidence on the mutual interactions between non-invasive brain stimulation and sex/polypeptide hormones. The PubMed database was searched from its inception to 31 March 2023, for (i) studies that investigated the impact of sex and/or polypeptide hormones on the effects induced by non-invasive brain stimulation, or (ii) studies that investigated non-invasive brain stimulation in the modulation of sex and/or polypeptide hormones. Eighteen studies (319 healthy and 96 disabled participants) were included. Most studies focused on female sex hormone levels during the menstrual cycle. The later follicular phase is associated with a weak between hemispheric and intracortical inhibition, strong intracortical facilitation, and high stimulation-induced neural and behavioural changes. The opposite effects are observed during the luteal phase. In addition, the participant’s sex, presence and/or absence of real ovulation and increase in oestradiol level by chorionic gonadotropin injection influence the stimulation-induced neurophysiological and behavioural effects. In Parkinson’s disease and consciousness disorders, the repetitive application of non-invasive brain stimulation increases oestradiol and dehydroepiandrosterone levels and reduces disability. To date, male hormones have not been sufficiently included in these studies. Here, we show that the sex and/or polypeptide hormones and non-invasive brain stimulation methods are in reciprocal interactions. This may be used to create a more effective and individualised approach for healthy individuals and individuals with disabilities.

## 1. Introduction

Non-invasive brain stimulation (NIBS) methods are innovative tools to investigate the functioning of neural networks and induce short- and long-term changes within the networks. However, the neural, behavioural, and cognitive effects induced by these methods vary significantly across individuals, with sex being one of the determining factors [1,2]. In addition to differences in the cortical bone structure and white matter, hormonal disparities are considered major generators of sex-related differences [2,3]. This study aimed to investigate the influence of sex and/or polypeptide hormones on NIBS-induced effects. In addition, existing evidence of opposing interactions (the influence of NIBS on sex and/or polypeptide hormones) was included and analysed. These hormones are essential components of several brain pathways [4] and may interact with neurotransmitters.

The following paragraphs focus on hormonal changes during the menstrual cycle and menopause in women because a major part of the existing evidence in this area is based on this.

Typical changes in hormone levels during the menstrual cycle are illustrated in Figure 1: the low hormonal level during menstruation, the peak of oestradiol, follicle-stimulating hormone, luteinizing hormone, and inhibin B levels before ovulation, and the culmination of progesterone and oestradiol during the luteal phase [5,6]. The “high hormone periods” are often associated with promoted neural recruitment, and superior behavioural, cognitive, and emotional abilities. For example, higher bilateral activation of related regions during semantic [7,8] and visuospatial [7] tasks has been detected during the mid-luteal phase than during the menstrual phase [7,8]. Better visuomotor performance [9] and higher muscular force in the lower limbs [10] were observed during the mid-luteal phase than during menstruation [9]. Ovulation is associated with greater lower limb muscle power than the late follicular and luteal phases [11]. The final decision regarding the effects of single hormones is difficult because of the overlap of hormonal peaks during the menstrual cycle [5,6]. Oestrogen and progesterone are considered to play crucial roles in these processes [4,12]; however, evidence of their effects is inconsistent. Enhanced brain recruitment may be associated with high or low levels of these hormones [8,13]. Higher levels of both oestradiol and progesterone correlate with greater brain region recruitment during cognitive tasks [8]. In contrast, contraception-induced inhibition of both hormones enhances neural activation and the performance of verbal tasks [13]. A previous review suggested the opposite effects of both hormones on the brain network, cognition, and emotions: an excitatory and inhibitory effect of oestradiol and progesterone, respectively [14].

Menopause is characterised by drop-in oestradiol, progesterone, and inhibin B, and an increase in follicle-stimulating hormone and luteinizing hormone levels [12,15]. It is associated with changes in brain structure, energy consumption, connectivity, and emotional and cognitive decline [16,17]. A magnetic resonance imaging/positron emission tomography (MRI/PET) study showed a significant loss of hippocampal volume and a decrease in glucose metabolism in several key brain regions in peri- and post-menopausal women [17]. This is associated with worsened memory and higher-order processing [17]. A review revealed a reduction of glucose metabolism in certain brain regions (hippocampus, entorhinal cortex, medial temporal cortex, and posterior cingulate) but an enhancement within other regions (pons, caudate, and praecuneus) through menopause [16]. Oestradiol and progesterone replacement may effectively counteract menopause-related neurophysiological changes and perverse cognitive abilities [16,18,19]. Women receiving either oestradiol or combined oestradiol–progesterone treatment had improved performance and higher neural activation during diverse cognitive tasks than those who did not receive hormonal therapy [18,19].

(Repetitive) transcranial magnetic stimulation ((r)TMS) and direct current stimulation (tDCS) are the most popular NIBS techniques. The following paragraphs describe the common (r)TMS and tDCS protocols used for (a) investigation or (b) modulation of neural networks.

Single- and paired-pulse transcranial magnetic stimulation (TMS) can provide information regarding neural networks in parallel with conventional functional magnetic resonance imaging (fMRI) and positron emission tomography (PET). Corticospinal excitability (motor-evoked potential amplitude and resting/active motor threshold intensity), cortical silent period (CSP), and ipsilateral silent period (ISP) are tested using single TMS pulses applied over the primary motor cortex (M1) [20]. The intracortical interactions (short-interval intracortical inhibition (SICI), long-interval intracortical inhibition (LICI), and intracortical facilitation (ICF)) are evaluated by paired TMS pulses applied time-shifted to one of the M1s [20]. The between- and inter-hemispheric interactions (inter-/intra-hemispheric inhibition/facilitation) are investigated using paired TMS pulses, which are applied time-shifted over two different brain regions using two coils [20]. The first (conditioning) coil is placed over any cortical region, and the second coil is placed over the ipsilateral or contralateral M1 [20]. The effects of single- and paired-pulse TMS show great inter- and intra-individual variability [21]. Sex and/or polypeptide hormones are several causes of this phenomenon.

Repetitive transcranial magnetic stimulation (rTMS) and direct current stimulation (tDCS) are NIBS methods that induce short- and long-term changes in neural networks outside the stimulation period [20,22]. Stimulation intensity and duration crucially determine the amount and preservation of their effects [20,22]. Modulation of cortical excitability beyond the stimulation period can be achieved by a 0.6 mA tDCS applied for 3 min. Short-term modulation of brain tissue (10–15 min) requires 1 mA tDCS for 5–7 min. For long-term modulation of corticospinal excitability (≤1 h), 1 mA tDCS over at least 11 min is needed [22]. A single session of rTMS modulates brain networks 30 min after stimulation completion [20]. The “direction” of NIBS-induced modulation is traditionally associated with rTMS frequencies and tDCS electrode polarity. High-frequency rTMS (≥5 Hz), intermittent theta burst stimulation (iTBS), paired-pulse rTMS (inter-stimulus interval, 1.5 ms), and anodal tDCS are considered to “facilitate” neural networks. In contrast, low-frequency rTMS (1 Hz), continuous theta burst stimulation (cTBS), paired-pulse rTMS (inter-stimulus interval, 3 ms), and cathodal tDCS are expected to induce their “inhibition” [20,22]. More recent research has demonstrated repeated NIBS-induced modulation outside this simplified framework [23,24,25,26]. Sex and/or polypeptide hormones may play an important role in this regard. They are crucially involved in several neural pathways and reciprocally interact with neural changes [4].

## 2. Materials and Methods

This systematic review aims to summarise and analyse the available evidence on non-invasive brain stimulation and sex/polypeptide hormones in reciprocal interactions. This study was conducted in accordance with the Preferred Reporting Items for Systematic Reviews and Meta-analysis (PRISMA) guidelines [27] without previous registration.

### 2.1. Search Strategy

The PubMed electronic database was searched from its inception to 31 January 2023, for randomised controlled and clinical trials by one reviewer. The combination of the following search terms was used: (1) “TMS” or “rTMS” or “tDCS” or “tACS” or “tsDCS” or “tsACS” or “non-invasive brain stimulation” and (2) “hormones” or “sex hormones” or “polypeptide hormones” or “oestrogen” or “oestradiol” or “progesterone” or “androgen” or “testosterone” or “luteinizing hormone” or “follicle-stimulating hormone”. The search process is illustrated in Figure 2.

### 2.2. Eligibility Criteria

Studies matching the following criteria were selected: (1) original prospective studies (observational or interventional); (2) human cohort; (3) five participants or more; (4) sex hormones (androgen, oestrogen, progesterone and testosterone) and/or polypeptide hormones (luteinizing hormone, follicle-stimulating hormone, and gonadotropin-releasing hormone) and/or relevant precursors (dehydroepiandrosterone (DHEA)) evaluation; (5) non-invasive brain stimulation (TMS, rTMS, tDCS, tACS, tsDCS and tsACS) application, and (6) English or German language.

### 2.3. Data Extraction

The primary data extracted were (1) the level of sex and/or polypeptide hormones and/or relevant precursors (menstrual cycle day) and (2) the effects of NIBS on neural networks, cognition, behaviour and hormone levels. The objective evaluation of NIBS- and hormone-induced effects based on statistically significant/no-significant time*intervention interaction between real and sham, as reported in the manuscripts included. In addition, the participants’ characteristics (number, age, sex and health status) and the study design aspects (interventional versus observational, evaluation schedule, and stimulation parameters) were recorded.

### 2.4. Data Synthesis

The selected studies were divided into three groups, based on their research approach: (i) single- and double-TMS studies evaluating the influence of sex and/or polypeptide hormones and/or relevant precursors on neural networks; (ii) interventional experiments evaluating the modulatory influence of hormones on rTMS-, tDCS-, tACS-, tsDCS- and tsACS-induced effects; and (iii) interventional experiments investigating the potential of rTMS, tDCS, tACS, tsDCS and tsACS in the modulation of sex and/or polypeptide hormones.

### 2.5. Methodological Quality Assessment

The methodological quality of the studies, such as random allocation, probands´ and assessors´ blinding, dropout rate, etc., was evaluated (if applicable) using an 11-item PEDro scale [28]. The higher the total score, the better the methodological quality (10–9, excellent; 8–6, good; 5–4, fair; and <4, poor).

## 3. Results

### 3.1. Single- and Paired-Pulse Tms Studies That Evaluated the Influence of Hormones on Neural Networks

Overall, eight single- and/or paired-pulse TMS experiments investigated the influence of sex hormones and/or polypeptide hormones on human neural networks [29,30,31,32,33,34,35,36]. Table 1 provides an overview of these studies.

#### 3.1.1. Study Design and Cohorts

Seven studies tested healthy women during different phases of their menstrual cycle [29,31,32,33,34,35,36]. Measurements were performed (a) during the later follicular phase/ovulation (D7-16) and (b) during the luteal phase (D18-27) in all studies. Additional evaluation during the early follicular phase/menstruation (D0-9) was performed in five studies [29,32,33,34,36]. Two studies included healthy men as a control group [31,36]. One study tested the effects of a single chorionic gonadotropin injection over 21 days in healthy men [30]. A total of 121 women and 33 men with average ages of 21 ± 2 years and 34 ± 8 years, respectively, were included. All studies except two [30,31] were purely observational.

#### 3.1.2. Hormone Evaluations

In addition to blood tests [29,30,31,32,33], saliva [36], and urine [34] tests were performed to determine hormone levels during the menstrual cycle. The available data were highly inconsistent across studies regarding hormone levels in identical phases of the menstrual cycle, and hormone fluctuations across the menstrual cycle.

Oestradiol measurements varied between 42 ± 16 pg/mL and 248 ± 129 pg/mL during menstruation (D2-D4), between 64 ± 31 pg/mL and 328 ± 160 pg/mL during the later follicular phase/ovulation (D5-D16), and, between 44 ± 30 pg/mL and 341 ±186 pg/mL during the luteal phase (D18-24) [29,30,31,32,33,34]. One study showed an increase in oestradiol levels between menstruation and ovulation, and a decrease during the luteal phase [34]. In contrast, three studies showed a further increase in this hormone during the luteal phase [29,31,33]. A study showed that the oestradiol level in ovulatory (103 ± 58 pg/mL) women is higher than that in anovulatory women (41 ± 16 pg/mL) during the later follicular phase/ovulation [32]. The oestradiol levels in men had fewer variations (between 32 ± 8 pg/mL and 51 ± 17 pg/mL) across studies [30,31]. A human chorionic gonadotropin injection induced 155 ± 87 pg/mL of oestradiol within 2 days, which decreased to 44 ± 30 pg/mL after 16 days [31].

Progesterone levels ranged between 1.0 ± 0.7 ng/mL and 1.5 ± 0.6 ng/mL during menstruation (D2-4), 0.3 ± 0.1 ng/mL and 1.6 ± 0.8 ng/mL during later follicular phase/ovulation (D8-14), and 4.4 ± 4.6 ng/mL and 12 ± 5.0 ng/mL during luteal phase (D18-24) [29,31,32,33,34]. The data consistently showed a low level of progesterone during the first half of the menstrual cycle and a significant increase during the luteal phase [29,31,33,34]. Ovulatory women (12 ± 5.4 ng/mL) show significantly higher progesterone levels during the luteal phase than anovulatory women (0.7 ± 0.5 ng/mL) [32]. A single study detected the level of progesterone in men to be 0.4 ± 0.1 ng/mL [31].

One study measured luteinizing hormone levels, which were significantly higher in ovulatory women (9.7 ± 8.8 mIU/mL) than in anovulatory women (4.2 ± 1.1 mIU/mL) during the late follicular phase/ovulation (D11-16) [32].

Testosterone levels in women and their changes during the menstrual cycle were evaluated in a single study [31]. No significant differences were detected between the later follicular phase (0.3 ± 0.1 ng/mL on D8) and the luteal phase (0.04 ± 0.2 ng/mL on D22). Testosterone levels in men varied from 5.3 ± 1.2 ng/mL to 6.9 ± 2.8 ng/mL [30,31]. Its level increased to 12.1 ± 5.5 ng/mL 2 days after human chorionic gonadotropin injection and decreased to 7.4 ± 3.2 ng/mL 16 days later [31].

#### 3.1.3. Neural Network Evaluations

Resting motor threshold (rMT), active motor threshold (aMT), and/or motor-evoked potential (MEP) amplitude were evaluated in all included studies [29,30,31,32,33,34,35,36]. One study demonstrated that the MEP amplitude was greater in men than in women (independent of the menstrual cycle phase) [31]. Another study showed that an increase in oestradiol levels in men (after human chorionic gonadotropin injection) was associated with a decrease in rMT [30]. No significant effects were detected in the remaining studies.

Short-interval intracortical inhibition (ICI) was assessed in six studies [29,31,32,34,35,36]. One study demonstrated that the SICI is (1) stronger during the luteal phase (D21) and (2) weaker during ovulation (D14) than during menstruation (D2) (29). Another study indicated that SICI is stronger during both the luteal phase (D18-24) and menstruation (D4) than during the later follicular phase (D11) [34]. Similarly, another study reported stronger SICI during the luteal phase (D18-27) than during the later follicular phase [35]. One study indicated that the strength and variation of SICI during the menstrual cycle were dependent on the release of eggs from the ovaries [32]. Ovulatory women showed weaker SICI than anovulatory women during menstruation (D2), ovulation (D11-16), and the follicular phase (D18-23) [32]. Furthermore, only anovulatory women showed significant changes in SICI during the menstrual cycle: the strongest SICI during menstruation (D2) and a weaker SICI during the later follicular (D8) and luteal (D20) phases [32]. In addition, a correlation was detected between high oestradiol levels and weak SICI during ovulation (D11-16) [32]. Two studies did not detect any menstrual cycle-related SICI changes [31,36].

Intracortical facilitation (ICF) was determined in three studies [34,35,36]. One study showed that ICF was stronger during the later follicular phase (D7-12) than during the luteal phase (D18-27) [35]. Correspondingly, another study showed stronger ICF during the later follicular phase (D9-12) than during both the luteal phase (D19-25) and menstruation (D2-5) [34]. However, one study did not show hormone-dependent effects [36].

CSP was analysed in three studies [29,32,33]. None of the studies showed significant effects.

ISP was analysed in a previous study [33]. The ISP was significantly longer during the luteal phase (D21-22 with high levels of progesterone) than during the later follicular phase (D9 with high levels of oestradiol) [33]. Furthermore, (1) the higher the oestradiol level during the follicular phase and (2) the higher the progesterone level during the later luteal phase, the shorter the ISP during these phases [33].

The transcallosal conduction time (TCT) was analysed in one study, which showed no relevant effects [33].

#### 3.1.4. Further Evaluations

Three studies performed examinations in addition to neural networks and hormones examinations [31,34,36]. The data showed that hand motor function was significantly better during the luteal phase (D18-24) than during ovulation (D12-15) [36]. Neither menstrual cycle-dependent changes nor relationships with neural processing and/or hormones were detected for depression, anxiety, premenstrual syndrome [34] and the brain-derived neurotrophic factor or the insulin-like growth factor [31].

### 3.2. Interventional Studies Investigating the Influence of Hormones on rTMS- and tDCS-Induced Effects

Six studies evaluated the effects of NIBS under various sex hormone and/or polypeptide hormone conditions [37,38,39,40,41,42]. Table 2 provides an overview of these studies. Table 3 presents their methodological quality.

#### 3.2.1. Study Design, Cohorts, and Quality

Five studies evaluated the effects of rTMS [37,40] or tDCS [38,41,42] in healthy women during different phases of the menstrual cycle. A single intervention was applied during menstruation (D1-5) and the later follicular phase/ovulation (D6-16) in all studies. Two studies also applied an intervention during the luteal phase (D18-22) [38,42]. All trials except one [38] included healthy male controls. They were aged from 23 ± 4 years to 30 ± 2 years. One trial compared the effect of ten rTMS sessions on postmenopausal women (age, 58 ± 9 years) and men (age, 44 ± 15) who had depression [39]. The PEDro score varied between three (poor methodological quality) and seven (good methodological quality).

#### 3.2.2. Hormones Evaluations

All studies on healthy cohorts performed blood tests to determine hormone levels. However, the data were inconsistent.

The oestradiol levels ranged between 35 ± 8 pg/mL and 49 ± 25 pg/mL during menstruation (D1-5), and 111 ± 52 pg/mL and 236 ± 16 pg/mL during the later follicular phase/ovulation (D6-16). A single study evaluated oestradiol levels during the luteal phase and did not observe relevant differences from those observed during the later follicular phase/ovulation [38]. The oestradiol levels in healthy men varied between 25 ± 7 pg/mL and 27 ± 13 pg/mL. Middle-aged individuals with depression showed lower oestrogen levels (women, 20 ± 7 pg/mL; men, 12 ± 4 pg/mL) than healthy, younger individuals [39].

One study detected that progesterone levels in healthy women ranged from 0.7 ± 0.2 ng/mL during menstruation (D1) to 2.4 ± 0.4 ng/mL during the later follicular phase/ovulation (D14) [40]. Individual patients with depression demonstrated a lower value of progesterone (women, 0.02 ± 0.02 ng/mL; men, 0.03 ± 0.01 ng/mL) than younger healthy individuals [39].

Luteinizing hormone levels were determined in a single study involving individuals with depression [39]. Women aged 58 ± 9 years (43 ± 29 mIU/mL) showed lower values of luteinizing hormone than men 14 years younger (4.0 ± 1.9 mIU/mL).

Follicle-stimulating hormone levels in individuals with depression were measured in a single study [39]. Women (36 ± 13 mIU/mL) demonstrated higher levels of follicle-stimulation hormone than men (3.0 ± 2.0 mIU/mL).

#### 3.2.3. Neural Networks Evaluation

rMT and/or MEP amplitudes were measured in a single study [40]. A 5Hz rTMS over the left M1 induced an increase in MEP amplitude only during ovulation (D14), and not during menstruation (D1) in women. In contrast, an increase in the MEP amplitude was observed on both examination days in men. No effects were detected in rMT [40].

CSP was investigated in a single study [40]. A 5Hz rTMS over the left M1 led to an increase in both women during menstruation (D1) and ovulation (D14), as well as in men [40].

TMS-evoked potentials (TEPs) were recorded in two studies [37,41]. The 10 Hz rTMS over the left dorsolateral prefrontal cortex (DLPFC) decreased TEPs N45 and N100 and increased TEPs P60 during ovulation (D15-16). During menstruation (D2-5), a stimulation-induced decrease in TEPs P180 amplitude was detected [37]. Anodal 1 mA tDCS over the left DLPFC increased TEPs P200 amplitude during the later follicular phase (D6-9) but not during menstruation (D2-5) [41]. TEPs N120 was not influenced by the menstrual cycle [41]. No stimulation-induced changes were detected in men [37,41]. In addition, pre-intervention data have shown sex-related differences [37,41]. Women showed higher TEPs P180 [37] and TEPs P200 [41] during both menstruation (D2-5) [37,41] and ovulation (D15-16) than those in men [37].

#### 3.2.4. Further Evaluations

Three studies performed additional examinations apart from the neural network and hormone examinations [38,39,42]. Anodal 4mA tDCS over the left DLPFC applied during the later follicular (D9-10) and luteal (D18-20) phases but not during menstruation (D3-4) worsened muscular performance during knee extension [38]. In contrast, a stimulation-induced increase in electromyography activity within the right knee extensors was detected independent of the menstrual cycle. No significant effects were observed in the knee flexors [38]. A study showed that anodal 2 mA tDCS over the left DLPFC modulated visuospatial ability in men but not in women [42]. Similarly, 5 Hz rTMS over the left DLPFC reduced depression symptoms in men but not in women [39].

### 3.3. Interventional Studies Investigating the Influence of rTMS on Hormones Levels

Four placebo-controlled studies investigated whether NIBS could be effective in modulating sex and/or polypeptide hormones in healthy individuals [43,44] or individuals with disabilities [45,46]. Table 4 shows an overview of these studies. Table 3 presents their methodological quality.

#### 3.3.1. Study Design, Cohorts, and Quality

Two studies investigated the effects of a single rTMS session in healthy young cohorts [43,44]. Seventeen women and 19 men were enrolled in these studies. One study applied 20 Hz rTMS over the (a) DLPFC and (b) M1 of the dominant hemisphere [14]. The other combined 10 Hz and 20 Hz rTMS over the left DPMC with an intensity of (a) 95% rMT or (b) 110% rMT [44]. Two studies examined the effects of multiple rTMS sessions in patients with disabilities [45,46]. The first applied 20 sessions of 10 Hz rTMS bilaterally over the M1 in patients with Parkinson´s disease [45]. The second study performed 10 sessions of 10 Hz rTMS over the left DLPFC in men with consciousness disorder [44]. Blood [44,46] or saliva [43] samples were used to determine hormone levels. One study provided no data regarding this [45]. The PEDro score varied between 6 and 7. This indicates a good methodological quality of studies.

#### 3.3.2. Hormones and Further Evaluations

Next, 10Hz rTMS applied over 10 days evoked both increased oestradiol levels and improved consciousness in comatose men [44]. The stronger the increase in oestradiol level, the greater the awareness [44]. A total of 20 sessions of 10 Hz rTMS applied bilaterally over the M1 improved the clinical symptoms of patients with Parkinson´s disease [45]. A stimulation-induced increase in DHEA levels was observed only in men [45]. Similarly, pre-intervention data showed a negative correlation between DHEA levels and disability only in men [45]. No rTMS-induced changes in cortisol levels were detected [45]. A single session of subthreshold 10 Hz and 20 Hz rTMS over the left DLPFC decreased cortisol and thyroid-stimulating hormone levels, respectively, in healthy young adults [44]. However, suprathreshold stimulation did not induce any significant effects. Interestingly, the women showed lower prolactin levels after placebo stimulation. No significant effects were detected for follicle-stimulating hormones [44]. A 20 Hz rTMS applied over the DLPFC than M1 induced a greater reduction of anxiety in healthy young individuals [43]. No significant effects of either protocol were found on testosterone and cortisol levels, heart rate, mood, or motivation [43].

Collectively, the available data indicate that (1) the sex and/or polypeptide hormones influence the human neural network and behaviour, as well as the effects induced by rTMS and tDCS, and (2) NIBS can modulate the level of sex and/or polypeptide hormones and/or relevant precursors.

## 4. Discussion

To our knowledge, this is the first systematic review that analyses the reciprocal effects of sex/polypeptide hormones and NIBS. Our findings support the understanding of neural and hormonal control in healthy and disabled people, in parallel to neuroimaging studies, and contribute to the development of innovative therapeutic strategies in several cohorts. Now, we will discuss our findings and their implementation within the broader framework of available research.

A major part of analysed studies investigated women during different phases of the menstrual cycle. The single- and paired-pulse experiments show that neural processes fluctuate through the menstrual cycle, in parallel to hormonal changes (Table 4). An opposite brain state was demonstrated during the later follicular phase/ovulation (weak between hemispheric and intracortical inhibition and strong intracortical facilitation) and the luteal phase (strong between hemispheric and intracortical inhibition and weak intracortical facilitation). Similarly, a previous systematic review of neuroimaging data shows a different brain structure and functional activation during the later follicular phase in comparison to both, the early follicular phase, and the mild luteal phase [47]. For example, a later follicular phase was associated with a larger grey matter volume in the hippocampus, insula, and cerebellum. This correlated positively with oestrogen and negatively with progesterone levels [47]. In addition, late follicular and mild luteal phases were associated with a higher BOLD response in the hippocampus during affective and cognitive processing than early follicular and late luteal phases. The higher oestrogen level, the greater the brain reactivity [47]. During the mild luteal phase, a greater resting-state functional connectivity of the middle frontal gyrus with anterior cingulate cortex, the amygdala, and the fronto-parietal network was observed, which correlated with oestrogen level [47]. Thus, this systematic review [47] as well as our data indicates consistently, that oestradiol and progesterone can exert opposite effects on the brain network. Accordingly, an earlier review emphasises that oestradiol activates and progesterone depresses neural processes [14]. A more recent review reaches the conclusion that ovarian hormones (estradiol and progesterone) enhance cortico-cortical and subcortico-cortical functional connectivity, whereas androgens (testosterone) decrease subcortico-cortical functional connectivity but increase functional connectivity between subcortical brain areas [48].

Parallel to electrophysiological data, opposite hand motor performance during the late follicular phase (poor hand function) and luteal phase (favourable hand function) was detected in our study. Similarly, previous studies show that visuospatial and motor performance is most lateralised (towards the right side) during menstruation, and most symmetrical during the follicular [49,50] or luteal phases [51]. In contrast, a current systematic review found no effects of the menstrual cycle phase regarding motor performance and motor exercise-induced fatigability in a major part of experiments [52,53]. Another review indicates that only emotions, but not cognitive abilities are affected by the menstrual cycle phase [54].

Regarding NIBS-induced neurophysiological and behavioural effects, significant differences were detected between the early follicular phase/menstruation (low effectiveness) and the late luteal phase/ovulation (high effectiveness) in our study. More studies are needed that compare the effects of several NIBS protocols through the menstrual cycle to support the development of more targeted interventions in women. The investigations of psychiatric syndromes across the menstrual cycle have repeatedly demonstrated that the premenstrual and menstrual phases are often associated with symptom exacerbation in psychosis, mania, depression, suicide/suicide attempts, and alcohol use [55]. Anxiety, stress, and binge eating appear to be elevated more generally throughout the luteal phase [55]. The subjective effects of smoking and cocaine use are reduced during the luteal phase [55]. Menstrual cycle coupled with NIBS may be an innovative approach to support the recovery of these diseases. Future NIBS research should strongly focus on the luteal phase, as it is less investigated up to now.

Our study shows that in addition to the menstrual cycle phase, the presence or absence of real ovulation also affects brain functions. The absence of real ovulation during the menstrual cycle is one of the major causes of infertility in women [56]. This disorder is associated with lower levels of oestradiol and progesterone and (b) higher levels of follicle-stimulating and luteinizing hormones [57]. Our data shows a stronger intracortical inhibition in these women through all phases of the menstrual cycle. Similarly, a decreased amplitude of low-frequency fluctuation within the left middle frontal gyrus as well as its increased functional connectivity with the left inferior frontal gyrus were detected in anovulatory cohorts, in comparison to ovulatory controls during fMRI experiments [58]. The neural abnormalities correlated significantly with both, hormonal disparities as well as worsened emotional, behavioural, and cognitive status [58]. Similarly, women with polycystic ovary syndrome (another common cause of infertility) show not only hormonal imbalance and metabolism changes that may cause missed or irregular menstrual periods, but also brain function changes [58,59]. This is coupled with cognitive, emotional and behavioural decline. It is an open question if NIBS could be used to support the effects of conventional therapies [60,61,62] in these cohorts.

Our data repeatedly demonstrated that the sex of the participants influenced (r)TMS- and tDCS-induced effects. Lower corticospinal excitability [31] and higher cortical excitability [37,41] were detected in women than in men. Stimulation-induced changes (increase and decrease) in cortical excitability were detected only in women [37,41]. Increased corticospinal excitability [40], modulated visuospatial ability [42], and reduced depression were observed only in men [39]. Thus, neither men nor women showed superior modulation capabilities, despite the differences detected. In contrast, previous studies have suggested that differences in craniofacial anatomy and brain tissue volume may lead to a higher brain response in women, especially when TMS is delivered to the prefrontal cortex [2,3]. Similarly, ample evidence suggests a superior modulation capability in women due to higher oestradiol and progesterone levels [2,3,4,13,14,63,64]. The role of male hormones in the field was neglected in previous studies. Although, several data point to their key roles in neural processing [65,66]. Future research should focus on this topic.

A study demonstrated that a transient increase in gonadal steroids after chorionic gonadotropin injection in men led to an increase in oestradiol levels and a reduction of MT [30], which concord with previous observations. More studies are needed to prove if NIBS´ effects may be enhanced by hormone replacement. Huge data shows that oestradiol and progesterone administration counteract menopause-related neural and cognitive decline [16,18,19]. Additionally, this postmenopausal hormone therapy may delay the progression of Alzheimer’s disease, dementia, and Parkinson’s disease [67]. Sex hormones are core elements of several pathways within the brain (such as glutamatergic, gamma-aminobutyric acidergic, dopaminergic, and serotonergic pathways) [4] and interact reciprocally with neurotransmitters. The synergies between NIBS and hormone therapy may be used to create innovative rehabilitation approaches for several cohorts.

In accordance, our data shows that the repetitive application of NIBS increased oestradiol levels and improved consciousness in coma patients [46], as well as increased DHEA (precursors of androgenic and oestrogenic hormones) levels and reduced disability in patients with Parkinson´s disease [45].

## 5. Conclusions

Present data indicate that reciprocal interactions exist between sex hormones, the neural system, and NIBS-induced effects. However, the existing evidence is too small, and future studies are needed. Almost all existing studies focused on female sex hormones. There exists hardly any data regarding male sex hormones. More research in this field may enlarge the view of the neural-hormonal background of behaviour, cognition, and emotions in healthy and disabled people. This may support the development of more effective targeted therapies in several cohorts.

## 6. Limitations

The objective interpretation of the findings was complicated by the presence of few studies and inconsistent data regarding research questions (hormones modulating NIBS and NIBS influencing hormones), stimulation protocols (single-pulse, double-pulse, repetitive applications), outcomes (different neurophysiological and behavioural outcomes), and probands (healthy, disabled, and different age categories). Furthermore, our study demonstrated that menstrual cycle-related oestradiol fluctuations differ slightly from those described in previous studies [5,6]. Our study showed an increase in oestradiol during both the later follicular and luteal phases. In contrast, a decrease in this hormone during the luteal phase was described previously [5,6]. This hampered the interpretation of our data and the discussion of the background of previous studies

## Figures and Tables

**Figure 1 biomedicines-11-01981-f001:**
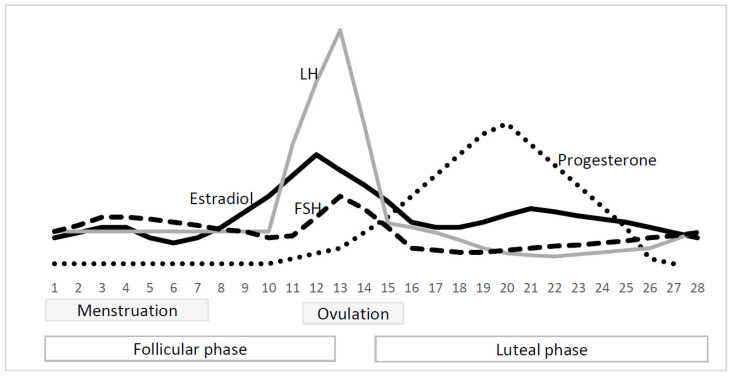
The changes in hormone levels during menstrual cycle [6]. Notes: FSH = follicle-stimulating hormone; LH = luteinizing hormone.

**Figure 2 biomedicines-11-01981-f002:**
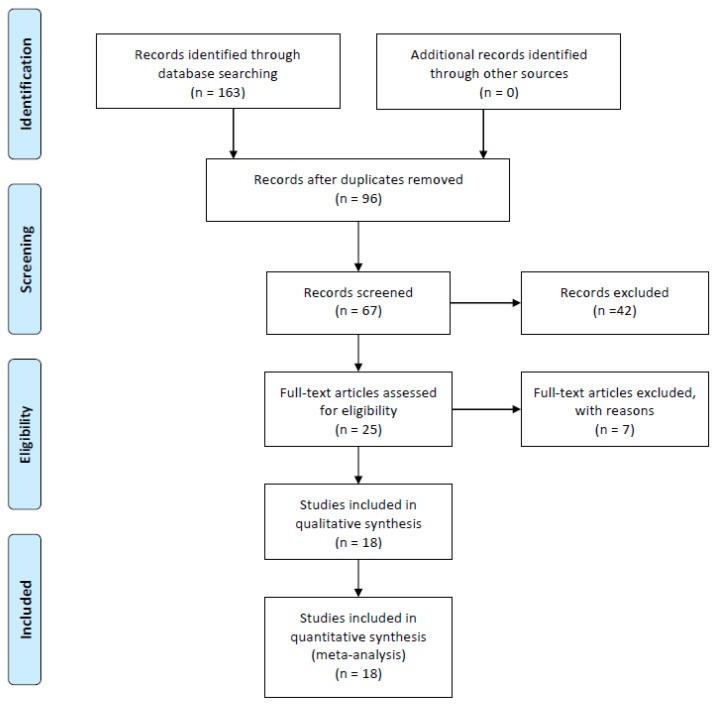
PRISMA flow diagram.

**Table 1 biomedicines-11-01981-t001:** Human studies investigating the influence of sex and/or polypeptide hormones on effects induced by single- and double-pulse TMS.

Reference	Participants’ Number, Gender, Age, Diagnosis	Study Design/Evaluations Schedule (Menstrual Cycle Day)	E2, (pg/mL), P4, T (ng/mL), LU (mIU/mL) Level/Hormone Testing Methods	Outcomes/Results
Ansdell et al., 2019 [29]	30 women; 23 ± 2 years; healthy	observational/D2, D14, D21	E2: 248 ± 129 (D2), 328 ± 160 (D14), 341 ± 186 (D21); P4: 1.3 ± 0.5 (D2), 1.4 ± 0.7 (D14), 4.4 ± 4.6 (D21)/blood	SICI—(1) stronger on D21 than on D2 and on D14, and (2) stronger on D2 than on D14
				CSP duration, MEP amplitude—no effects
Hattemer et al., 2007 [32]	12 ovulatory, 8 anovulatory women; 25 ± 4 years; healthy	observational/D2, D8, D11–16, D18–23	na (D2, D8); E2: 103 ± 58 ovulatory, 41 ± 16 anovulatory, LH 9.7 ± 8.8 ovulatory, 4.2 ± 1.1 anovulatory (D11–16); P4: 12 ± 5.4 ovulatory, 0.7 ± 0.5 anovulatory (D18–23)/blood	ICI—(1) stronger on D2 than on D8 and on D18–23 in anovulatory women, (2) stronger on D11–15 than on D8 in anovulatory women, (3) stronger in anovulatory women than in ovulatory women on D2, D11–16, D18–23, (4) strength correlates negatively with E2 level on D11–16
				rMT, MEP, CSP—no effects
Hausmann et al., 2006 [33]	13 women; 24 ± 4 years; healthy	observational/D2, D8–10, D21–22	E2: 42 ± 16 (D2), 64 ± 31 (D8–10), 117 ± 49 (D21–22); P4: 1.5 ± 0.6 (D2), 1.6 ± 0.8 (D8–10), 12 ± 5.0 (D21–22)/blood	ISP duration—(1) longer on D21–22 than on D8–10, (2) correlated negatively with E2 on D8–10, and P4 on D21–22
				MT, CSP duration, TCT duration—no effects
Schmith et al., 1999 [35]	13 women; 34 ± 8 years; healthy	observational/D7–12, D18–27	na (D7–12, D18–27)/na	SICI—stronger on D18–27 than on D7–12
				ICF—stronger on D7–12 than on D18–27
				rMT—no effects
Schmith et al., 2002 [34]	18 women; 33 ± 8 years; healthy	observational/D4, D11, D18–24	E2: 44 ± 29 (D4), 148 ± 78 (D11), 117 ± 62 (D18–24); P4: (1.0 ± 0.7 (D4), 1.1 ± 0.7 (D11), 11 ± 6 (D18–24)/urine	SICI—stronger on D2–5 and on D19–25 than on D9–12
				ICF—stronger on D9–12 than on D2–5 and D19–25
				rMT, aMT, depression (BDI), anxiety (STAI), premenstrual tension syndrome—no effects
Zoghi et al., 2015 [36]	10 women, 10 men; 26 ± 5 years; healthy	observational/(D0–9, D12–15, D18–24)	na (D0–9, D12–15, D18–24)/saliva	hand function (Grooved Pegboard Test)—better on D18–24 than on D12–15 in women
				rMT, ICI, ICF—no effects
El-Sayes et al., 2019 [31]	17 women, 17 men; 21 ± 2 years; healthy	interventional (cycling (20 min, 65–70% of HRmax); without control; pre, post evaluation/D8, D22	E2: 43 ± 16 (D8), 157 ± 53 (D22), men 32 ± 8; P4: 0.3 ± 0.1 (D8), 8.5 ± 6.0 (D22), men 0,4 ± 0.1; T: 0.3 ± 0.1 (D8), 0.4 ± 0.2 (D22), men 5.3 ± 1.2/blood	MEP amplitude—(1) greater in men than in women (pre), (2) increased in both sexes after cycling
				SICI—weaker in both sexes after cycling
				BDNF, IGF-1—no effects
Bonifazi et al., 2004 [30]	6 men; 31 (27–42) years; healthy	interventional (human chorionic gonadotropin (5000 IU) injection on D3); without control; D1, D5, D21	E2: 51 ± 17 (D1), 155 ± 87 (D5), 44 ± 30 (D21); T: 6.9 ± 2.8 (D1), 12.1 ± 5.5 (D5), 7.4 ± 3.2/blood	rMT—lower on D5 than on D1 and D21
				MEP amplitude, duration—no effects

Notes: BDI = Beck Depression Inventory; BDNF = brain-derived neurotrophic factor; CSP = contralateral silent period; D = day (in men)/menstrual cycle day (in women); E2 = oestradiol; HRmax = maximum heart rate; ICF = intracortical facilitation; ICI = intracortical inhibition; IGF-1 = insulin-like growth factor; ISP = ipsilateral silent period, LH = luteinizing hormone; mL = millilitre; (m)IU = (milli)-international unit; ng = nanogram; pg = picogram; P4 = progesterone; (r)MT = resting motor threshold; SICI = short-interval cortical inhibition; STAI = State-trait Anxiety Inventory; TCT = transcallosal conduction time.

**Table 2 biomedicines-11-01981-t002:** Human studies investigating the influence of sex and/or polypeptide hormones on effects induced by rTMS and tDCS.

Reference	Participants’ Number, Gender, Age, Diagnosis	Study Design/Sessions Number/Evaluations Schedule/(Menstrual Cycle) Day	E2 (pg/mL), P4 (ng/mL), DHEA (μmol/L), FSH, LU (mIU/mL) Level/Hormone Testing Methods	Intervention	Outcomes/Results
Chung et al., 2018 [37]	14 women, 15 men; 25 ± 5 years; healthy	interventional; without placebo control; 1 session; pre, post 1, post 2 (20 min) evaluation/D2–5, D15–16	E2: 36 ± 11 (D2–5), 111 ± 52 (D15–16), men 25 ± 7/blood	10Hz rTMS (2300 pulses, 120% of rMT) over left DLPFC	TEPs P180 amplitude—(1) higher in women at pre, (2) decreased only in women on D2–5 (post 1), and D15–16 (post 1,2) after rTMS
					TEPs N100 amplitude—decreased only in women on D15–16 (post 1,2) after rTMS
					TEPs N45 amplitude—decreased only in women on D15–16 (post 1) after rTMS
					TEPs P60 amplitude—increased only in women on D15–16 (post 1) after rTMS
Inghilleri et al., 2004 [40]	8 women, 8 men; 30 ± 2 years; healthy	interventional; without placebo-control; 1 session; pre, post evaluation/D1, D14	E2: 35 ± 8 (D1), 236 ± 16 (D14); P4: 0.7 ± 0.2 (D1), 2.4 ± 0.4 (D14), men na/blood	5Hz rTMS (80 pulses, 120% of rMT) over left M1	MEP amplitude—(1) increased only in men at D1 after rTMS (2) increased in both sexes on D14 after rTMS, (3) correlated positively with estrogen in women on D14
					CSP duration—increased in both sexes on D1 and D14 after rTMS
					MT—no effects
Huang et al., 2008 [39]	14 women, 58 ± 9 years; 16 men, 44 ± 15 years; depression	interventional, without placebo-control; 10 sessions; pre, post evaluation/na	E2: women 20 ± 7, men 12 ± 4; P4: women 0.02 ± 0.02, men 0.03 ± 0.01; FSH: 36 ± 13, men 3 ± 2; LH: 43 ± 29, men 4.0 ± 1.9/blood	5Hz rTMS (1600 pulses, 100% of rMT) over left DLPFC	depression symptoms (HAM-D, BDI, CGI-S)—improved only in men after rTMS
Deters et al., 2022 [38]	10 women, = 24.3 ± 5.5 years; healthy	interventional; placebo-controlled; crossover; 1 session, post evaluation/D3–4, D9–10, D18–20	E2: 37 ± 17 (D3–4), 147 ± 107 (D9–10), 147 ± 10 (D18–20)/blood	anodal tDCS (20 min, 4mA) over left M1	fatigability (right, left) and torque (right) during knee extension—worsened on D9–10 and D18–20 after real tDCS
					EMG activity (right) knee extensors—increased after real tDCS
					fatigability and torque (right, left) during knee flexion, torque (left) during knee extension, EMG activity (right, left) knee flexors and (left) knee extensors—no effects
Lee et al., 2018 [41]	14 women, 15 men; 23 ± 4 years; healthy	interventional; without placebo-controlled; 1 session; pre, post 1, post 2 (20 min) evaluation/D2–5, D6–9	E2: 49 ± 25 (D2–5), 155 ± 53 (D6–9), men 27 ± 13/blood	anodal tDCS (15 min, 1mA) over left DLPFC	TEPs P200 amplitude—(1) higher in women on D2–5 than in men (pre), (2) increased only in women on D6–9 after tDCS (post 1,2)
					TEPs N120 amplitude—no effects
de Tomasso et al., 2014 [42]	10 women, 10 men; 26 ± 4 years; healthy	interventional; placebo-controlled; crossover; 1 session; pre, post evaluation/D1–3, D8–10, D21–22	na	anodal tDCS (20 min, 2mA) over left PC -P3	LBT 1 (1) greater left shift in men than in women (pre), (2) reduction of right shift in men after real tDCS
					LBT 2—no effects

Notes: BDI = Beck Depression Inventory; CGI-S = Clinical Global Impressions—severity of illness; CSP = contralateral silent period; D = menstrual cycle day; DLPFC = dorsolateral prefrontal cortex; EMG = electromyography; E2 = oestradiol; FSH = follicle-stimulating hormone; HAM-D = Hamilton Rating Scale for Depression; L = litre; LBT = line bisection test; LH = luteinizing hormone; min = minute; mL = millilitre; mIU = milli-international unit; MT = motor threshold, μmol = micromol; PC = parietal cortex; pg = picogram; P4 = progesterone; TEPs = TMS-evoked potentials.

**Table 3 biomedicines-11-01981-t003:** Human studies investigating the influence of non-invasive brain stimulation on sex and/or polypeptide hormone levels.

Reference	Participants’ Number, Gender, Age, Diagnosis	Intervention	Study Design	Outcomes/Results
Crewther et al., 2022 [43]	5 women, 8 men; 31 ± 9 years; healthy	20Hz rTMS (250 pulses, 90% of rMT over (a) DLPFC and (b) M1	interventional; placebo controlled; crossover; 1 session; pre, post 1, post 2 (15 min), post 3 (30 min) evaluation/saliva	anxiety (STAI)—decreased after rTMS over DLPFC in comparison to rTMS over M1
				T, cortisol, HR, mood (TMDS), motivation—no effects
Evers et al., 2000 [44]	12 women, 11 men; 32 ± 6 years; healthy	(a) 10Hz rTMS + 20Hz rTMS (810 pulses, 95% of rMT) over left DLPFC; (b) 10Hz rTMS + 20Hz rTMS (810 pulses, 110% of rMT) over left DLPFC	interventional; placebo controlled; crossover; 1 session; pre, post evaluation/blood	cortisol, TSH—decreased only after real rTMS at 95% of rMT
				prolactin—decreased only in women after placebo rTMS
				FSH—no effects
Aftanas et al., 2022 [45]	25 women, 21 men; 64 ± 2 years; Parkinson´s disease	10Hz rTMS over both M1 (4000 pulses, 100% of rMT) followed by 10Hz rTMS (3000 pulses, 110% of rMT) over left DLPFC	interventional; placebo controlled; parallel groups; 20 sessions; pre, post evaluation/na	DHEA—(1) increased only in men after real rTMS, (2) correlates only in men negatively with disability (pre)
				cortisol—no effects
				disability (UPRDS)—reduced after real rTMS in both sexes
He et al., 2020 [46]	50 men; 52 ± 2 years; consciousness disorders	10Hz rTMS (1000 pulses, 100% of rMT) over left DLPFC	interventional; placebo controlled; parallel groups; 10 sessions; pre, post evaluation/blood	estradiol—increased only after real rTMS
				consciousness (CRS-R)—(1) improved only after rel rTMS, (2) improvement correlated with increased estradiol

Notes: CRS-R = Coma Recovery Scale-revised; DHEA = dehydroepiandrosterone; FSH = follicle-stimulating hormone; HR = HR = heart rate; min = minute; STAI = State-trait Anxiety Inventory; T = testosterone; TMDS = total mood disturbance score; TSH = thyroid-stimulating hormone; UPRDS = Unified Parkinson´s Disease Rating Scale.

**Table 4 biomedicines-11-01981-t004:** Hormones, (r)TMS- and tDCS-induced effects, and motor function during the menstrual cycle.

		Early Follicular Phase/ Menstruation	Later Follicular Phase/Ovulation	Luteal Phase
Hormone levels				
	Estradiol	+	+++	++(+)
	Progesterone	+	+(+)	+++
	Follicle-stimulating hormone	+	+++	+
	Luteinizing hormone	+	+++	+

Single- and double-pulse TMS measurements				
	between hemispheric inhibition	++	+	+++
	intracortical inhibition	++	+	+++
	intracortical facilitation	+	+++	+

rTMS/tDCS-induced effects				
	motor function	+	+++	
	corticospinal excitability	+	+++	++
	TMS-evoked potentials	+	+++	

motor function		++	+	++

Notes: + = low/weak; ++ = moderate; +++ = high/strong.

## Data Availability

The datasets generated during and/or analysed during the current study are available from the corresponding author upon reasonable request.

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
