# Peer review of "Non-Invasive Brain Stimulation and Sex/Polypeptide Hormones in Reciprocal Interactions: A Systematic Review"

_biomedicines, 2023, doi:10.3390/biomedicines11071981_

Round 1

Reviewer 1 Report

The present review by Veldema entitled ‘Non-invasive brain stimulation and sex/polypeptide hormones in reciprocal interactions: A systematic review’ aimed to summarize and analyze the existing evidence on the reciprocal effects of sex/polypeptide hormones and non-invasive brain stimulation (NIBS). The review included 18 studies and focused on understanding how these hormones influence the human neural network and behavior, as well as the effects induced by NIBS. The findings suggest that sex and/or polypeptide hormones have an impact on neural networks, motor performance, and the effects of NIBS. Additionally, NIBS was found to modulate the levels of sex and/or polypeptide hormones and relevant precursors. The review also explored the influence of menstrual cycle phases and other hormone-related factors on the effects of NIBS. Overall, this review provides valuable insights into the associations between sex/polypeptide hormones and NIBS effects, emphasizing the need for more comprehensive research and considering individual hormonal profiles for optimizing NIBS interventions.

In general, I think the idea of this review is really interesting and the authors’ fascinating observations on this timely topic may be of interest to the readers of Biomedicines. However, some comments, as well as some crucial evidence that should be included to support the author’s argumentation, needed to be addressed to improve the quality of the manuscript, its adequacy, and its readability prior to the publication in the present form. My overall judgment is to publish this paper after the authors have carefully considered my suggestions below, in particular reshaping parts of the ‘Introduction’ and ‘Methods’ sections by adding more evidence.

Please consider the following comments:

According to the Journals’ guidelines, I would suggest the Authors to use the Biomedicines Microsoft Word template file.

Abstract: According to the Journal’s guidelines, the abstract should be a total of about 200 words maximum. And should be presented as a single paragraph, without subheadings. Please correct the actual one. Also, in my opinion, Authors should consider rephrasing this section. According to the Journal’s guidelines, the Abstract should contain most of the following kinds of information in brief form. Please, consider giving a more synthetic overview of the paper's key points: I would suggest rephrasing the results and conclusion to make them clear for readers to understand.

A graphical abstract that will visually summarize the main findings of the manuscript is highly recommended.

The objectives of this study are generally clear and to the point; however, I believe that there are some ambiguous points that require clarification or refining. I think that authors here need to be explicit regarding how they operationally investigated how sex and/or polypeptide hormones influence the human neural network and behavior, as well as the effects induced by NIBS, since this is the key aim of this review.

I would ask the Authors to better describe the criteria they decided to use for studies’ collection in their review: they should specify the requirements used to decide whether a study met the inclusion/exclusion criteria of the review, describe whether they included a balanced coverage of all information that is actually available, whether they have included the most recent and relevant studies and enough material to show the development and limitations in this field of interest. Finally, I believe that they should briefly present results of all statistical syntheses conducted.

Introduction: This section provides a good overview of the topic and the motivation for the study. However, it would be helpful to provide a brief explanation of the different NIBS methods mentioned (e.g., TMS, rTMS, tDCS) for readers who may not be familiar with these techniques. For this reason, I would suggest adding evidence from different studies that have explored (https://doi.org/10.1016/j.cub.2020.06.091; https://doi.org/10.1016/j.jad.2021.02.076). 

Methods: Consider providing more information about the search strategy, such as the specific keywords used and any filters or limitations applied during the search process. Furthermore, since this is a systematic review, it would be helpful to mention the criteria used for assessing the quality and risk of bias in the included studies.

Results: It would be useful to present the results in a more organized and structured manner. For example, you could group the studies based on the specific hormones investigated or the neural effects observed. Also, I would suggest to provide more details on the methods used to assess hormone levels and neural network evaluations in the included studies. 

Discussion: While this section mentions associations between sex/polypeptide hormones and NIBS effects, I recommend the Authors to clarify the nature of these relationships. Are the hormones directly causing the observed effects or are they simply correlated? Discuss the potential mechanisms underlying these relationships and consider alternative explanations for the findings, for example exploring how sex/polypeptide hormones influence the levels of these metabolites and their subsequent impact on neurotransmitter systems can provide a better understanding of the hormonal modulation of NIBS effects (DOI: 10.3390/biomedicines11030945; DOI: 10.3390/ijms24044114).

While the discussion briefly mentions the need for future research, it would be helpful to provide more specific recommendations for future studies. For this reason, I suggest including a proper ‘Limitations and future directions’ section to address key gaps in the current literature and propose research designs or methodologies that could address these limitations. 

References: Authors should consider revising the bibliography, as there are several incorrect citations. Indeed, according to the Journal’s guidelines, they should provide the abbreviated journal name in italics, the year of publication in bold, the volume number in italics for all the references. Also, some of the references are out of date:  please cite references from the last 10 years, particularly references from the recent 5 years.

Finally, what is the take-away message from this review article? It ends rather abruptly with no summary, no suggested directions or immediate challenges to overcome, no call to action, no indications of things we should stop trying, and only brief mention of alternative perspectives. What do the authors want us to take away from this paper?

Overall, I suggest submitting your work to an English native speaker to help with some grammar mistakes that can be found in different sections of the manuscript.

I hope that, after these careful revisions, this paper can meet the Journal’s high standards for publication. 

I am available for a new round of revision of this article. 

Best regards,

Reviewer

Minor editing of English language required.

Author Response

Dear Reviewer,

I appreciate the interest that you have taken in my manuscript and the constructive criticism you have given. I have revised our manuscript in accordance with them and have included a point-by-point response to your comments. The changes are marked in red.

Sincerely yours,

Jitka Veldema

Comment 1:

According to the Journals’ guidelines, I would suggest the Authors to use the Biomedicines Microsoft Word template file.

Answer: We revised the manuscript in accordance with this. The PRISMA checklist is attached.

Comment 2:

Abstract: According to the Journal’s guidelines, the abstract should be a total of about 200 words maximum. And should be presented as a single paragraph, without subheadings. Please correct the actual one. Also, in my opinion, Authors should consider rephrasing this section. According to the Journal’s guidelines, the Abstract should contain most of the following kinds of information in brief form. Please, consider giving a more synthetic overview of the paper's key points: I would suggest rephrasing the results and conclusion to make them clear for readers to understand.

Answer: Thank you for this comprehensive hint. We revised the abstract in accordance with them.

Comment 3:

A graphical abstract that will visually summarize the main findings of the manuscript is highly recommended.

Answer: We created a graphical abstract.

Comment 4:

The objectives of this study are generally clear and to the point; however, I believe that there are some ambiguous points that require clarification or refining. I think that authors here need to be explicit regarding how they operationally investigated how sex and/or polypeptide hormones influence the human neural network and behavior, as well as the effects induced by NIBS, since this is the key aim of this review.

Answer: The objective evaluation of NIBS- and hormone-induced effects bases on statistically significant/no-significant time*intervention interaction between real and sham, as reported in the manuscripts included. This information was included in the chapter Methods.

Comment 5:

I would ask the Authors to better describe the criteria they decided to use for studies’ collection in their review: they should specify the requirements used to decide whether a study met the inclusion/exclusion criteria of the review, describe whether they included a balanced coverage of all information that is actually available, whether they have included the most recent and relevant studies and enough material to show the development and limitations in this field of interest. Finally, I believe that they should briefly present results of all statistical syntheses conducted.

Answer: The inclusion/exclusion criteria as well as other relevant information regarding cohorts and methods are described in an appropriate manner in the chapter Eligibility criteria. Their presentation / discussion in included in the chapters Results, Discussion and Conclusions. The data synthesis is described in the chapter “Data synthesis” The PubMed database was searched from its inception to 31 March 2023 (as mentioned in Abstract and Methods) and all appropriate studies are included) A meta-analysis was not applied, because only a part of studies has an appropriate design.

Comment 6:

Introduction: This section provides a good overview of the topic and the motivation for the study. However, it would be helpful to provide a brief explanation of the different NIBS methods mentioned (e.g., TMS, rTMS, tDCS) for readers who may not be familiar with these techniques. For this reason, I would suggest adding evidence from different studies that have explored (https://doi.org/10.1016/j.cub.2020.06.091; https://doi.org/10.1016/j.jad.2021.02.076).

Answer: The NIBS methods are detailed described in the Introduction chapter

Comment 7:

Methods: Consider providing more information about the search strategy, such as the specific keywords used and any filters or limitations applied during the search process. Furthermore, since this is a systematic review, it would be helpful to mention the criteria used for assessing the quality and risk of bias in the included studies.

Answer: The information is included in the Methods chapter

Comment 8:

Results: It would be useful to present the results in a more organized and structured manner. For example, you could group the studies based on the specific hormones investigated or the neural effects observed. Also, I would suggest to provide more details on the methods used to assess hormone levels and neural network evaluations in the included studies.

Answer: The studies grouping bases on study design. In this framework, the data for specific hormones, the methods of their evaluation as well as neural effects observed are presented in a structured manner. The chapter Introduction contains a detailed description of neural networks evaluation methods.

Comment 9:

Discussion: While this section mentions associations between sex/polypeptide hormones and NIBS effects, I recommend the Authors to clarify the nature of these relationships. Are the hormones directly causing the observed effects or are they simply correlated? Discuss the potential mechanisms underlying these relationships and consider alternative explanations for the findings, for example exploring how sex/polypeptide hormones influence the levels of these metabolites and their subsequent impact on neurotransmitter systems can provide a better understanding of the hormonal modulation of NIBS effects (DOI: 10.3390/biomedicines11030945; DOI: 10.3390/ijms24044114).

Answer: Thank you for this hint. The discussion chapter was evaluated substantially in accordance with this comment.

Comment 10:

While the discussion briefly mentions the need for future research, it would be helpful to provide more specific recommendations for future studies. For this reason, I suggest including a proper ‘Limitations and future directions’ section to address key gaps in the current literature and propose research designs or methodologies that could address these limitations.

Answer: The chapter Conclusions (mentions the need for future research) was included, additional to chapter limitations.

Comment 11:

References: Authors should consider revising the bibliography, as there are several incorrect citations. Indeed, according to the Journal’s guidelines, they should provide the abbreviated journal name in italics, the year of publication in bold, the volume number in italics for all the references. Also, some of the references are out of date:  please cite references from the last 10 years, particularly references from the recent 5 years.

Answer: Only 18 appropriate studies were identified that match inclusion criteria. Thus, it is not reasonable to exclude references that are more than 10 / 5 years old. Furthermore, only ¼ of references is older than 10 years. The references formatting was revised in accordance with Journal’s guidelines. The references were checked for correctness.

Comment 11:

Finally, what is the take-away message from this review article? It ends rather abruptly with no summary, no suggested directions or immediate challenges to overcome, no call to action, no indications of things we should stop trying, and only brief mention of alternative perspectives. What do the authors want us to take away from this paper?

Answer: The information is included in the chapters discussion (has been revised substantially) and conclusion (was additionally included).

Comment 12:

Overall, I suggest submitting your work to an English native speaker to help with some grammar mistakes that can be found in different sections of the manuscript.

Answer: The first version of the draft was already revised by a language editing service.

Reviewer 2 Report

This review article focused on the interactions between various non-invasive brain stimulation techniques and sex/polpeptide hormones. The review ended up including 18 studies with over 400 total subjects. The main findings were that the majority of studies have involved female sex hormone levels during the menstrual cycle. Specifically, the later follicular phase was associated with weak between hemispheric and intracortical inhibition strong intracortical facilitation and high stimulation-induced neural and behavioural changes. In contrast, the luteal phase displayed the opposite patterns of results. Finally, biological sex, real ovulation, and increase in oestradiol level by chorionic gonadotropin injection influences stimulation-induced neurophysiological and behavioural effects.

Overall, I think this review was well-done, on an important topic, and provided a great deal of interesting information on the above topics. This topic is understudied in the field of brain stimulation in my opinion and the reasons for the wide range of responses to brain stimulation is an important issue in the field. Hormonal changes and levels could be one reason for this. Therefore, this review is timely and my lead to increased research in this area. The review was well-written and laid out overall and did not have any fatal flaws. I think it will be of interest to readers of the journal and adds to the literature on the topic. Thus, I think the paper should eventually be published and I only have a set of minor comments related to the writing, grammar, and organization of the paper. I believe the resolution of these issues will make this review better and more valuable. These minor issues are listed below.

1.      Section 1.3. I would refer to it as paired-pulse TMS instead of double pulse.

2.      Section 1.3, third line, there are too many parantheses around fMRI.

3.      Section 1.3, 4th line and throughout the paper, it should be referred to as resting motor threshold or active motor threshold when appropriate, not just motor threshold.

4.      Section 1.3, 5th line, I don’t think I have ever heard CSP and ISP referred to as long-lasting cortical inhibition. Same thing further down with IHI and IHF.

5.      Section 1.3, 8th line, write out these inhibitory and facilitatory pathways specifically here and throughout the paper. Write out the full name and then the abbreviations of SICI, LICI, SICF, and ICF, which I think you are referring to.

6.      Section 1.4 title. I think stimulations should read “stimulation”

7.      Section 1.4, line 6 and elsewhere. Use a different word than processing.

8.      Intracortical Inhibition (ICI section). I think you are referring to SICI which is sometimes just called ICI. It is much more common nowadays to call it SICI. Do this throughout.

9.      In the CSP, TCT sections there needs to be more info included than just “showed no relevant effects” for example. Some info and studies have to be briefly described in these sections.

10.  Some places in the manuscript the brain area stimulated in a study is not given. Recheck all these.

11.  Section 3.3.2 Can a more elaborate title be used than simply “Effects”?

12.  The bibliography has mistakes. Some study titles are not capitalized and others have all the words capitalized. As just one example compare references 14 and 16 to 15 and 17 in the format of the titles of the studies. Fix all of these in the bibliography.

13.  Overall the paper was well-written. There are some typos as mentioned above in the extra parentheses example. There are others like this in the paper but I don’t have time to point them all out. Please proof the paper again.

See my comments to authors

Author Response

Dear Reviewer,

I appreciate the interest that you have taken in my manuscript and the constructive criticism you have given. I have revised our manuscript in accordance with them and have included a point-by-point response to your comments. The changes are marked in red.

Sincerely yours,

Jitka Veldema

Comment 1:

Section 1.3. I would refer to it as paired-pulse TMS instead of double pulse.

Answer: Thank you for this hint. We revised the term.

Comment 2:

Section 1.3, third line, there are too many parantheses around fMRI.

Answer: We revised this.

Comment 3:

Section 1.3, 4th line and throughout the paper, it should be referred to as resting motor threshold or active motor threshold when appropriate, not just motor threshold.

Answer: We revised the term throughout the main text and the tables.

Comment 4:

Section 1.3, 5th line, I don’t think I have ever heard CSP and ISP referred to as long-lasting cortical inhibition. Same thing further down with IHI and IHF.

Answer: Thank you for the important hint. We removed the inaccurate descriptions and hope that the revised version is correct.

Comment 5:

Section 1.3, 8th line, write out these inhibitory and facilitatory pathways specifically here and throughout the paper. Write out the full name and then the abbreviations of SICI, LICI, SICF, and ICF, which I think you are referring to.

Answer: We revised the sentence as suggested

Comment 6:

Section 1.4 title. I think stimulations should read “stimulation”

Answer: We removed the titles in this chapter completely.

Comment 7:

Section 1.4, line 6 and elsewhere. Use a different word than processing.

Answer: We replaced the term “processing” with “processes”, “functions”, “tissue” throughout the manuscript.

Comment 8:

Intracortical Inhibition (ICI section). I think you are referring to SICI which is sometimes just called ICI. It is much more common nowadays to call it SICI. Do this throughout.

Answer: Thank you for this hint. We revised the manuscript and the Tables in accordance with them.

Comment 9:

In the CSP, TCT sections there needs to be more info included than just “showed no relevant effects” for example. Some info and studies have to be briefly described in these sections.

Answer: Both subsections show, in accordance with the remaining manuscript, the number of studies and the intervention-induced effects. If no significant differences between different phases of menstrual cycle were detected, there is nothing to report.

Comment 10:

Some places in the manuscript the brain area stimulated in a study is not given. Recheck all these.

Answer: We added this information.

Comment 11:

Section 3.3.2 Can a more elaborate title be used than simply “Effects”?

Answer: The tithe was renamed to “Hormones and further evaluations”.

Comment 12:

The bibliography has mistakes. Some study titles are not capitalized and others have all the words capitalized. As just one example compare references 14 and 16 to 15 and 17 in the format of the titles of the studies. Fix all of these in the bibliography.

Answer: Thank you for this hint. We revised the bibliography carefully.

Comment 13:

Overall the paper was well-written. There are some typos as mentioned above in the extra parentheses example. There are others like this in the paper but I don’t have time to point them all out. Please proof the paper again.

Answer: We revised the manuscript carefully for typos.

Round 2

Reviewer 1 Report

The authors did an excellent job clarifying all the questions I have raised in my previous round of review. Currently, this paper is a well-written, timely piece of research and provides a useful description of the existing evidence on the reciprocal effects of sex/polypeptide hormones and non-invasive brain stimulation (NIBS). 

Overall, this is a timely and needed work. It is well researched and nicely written, with a good balance between descriptive and narrative text.

I believe that this paper does not need a further revision, therefore the manuscript meets the Journal’s high standards for publication.

I am always available for other reviews of such interesting and important articles.

Reviewer